# A Refractive Index-Based Dual-Band Metamaterial Sensor Design and Analysis for Biomedical Sensing Applications

**DOI:** 10.3390/s25010232

**Published:** 2025-01-03

**Authors:** Lakshmi Darsi, Goutam Rana

**Affiliations:** Department of Electronics and Communication Engineering, SRM University, Guntur 522240, Andhra Pradesh, India; lakshmi_darsi@srmap.edu.in

**Keywords:** metamaterial sensor, biomedical sensing, non-destructive in vitro sensing, high sensitivity

## Abstract

We propose herein a metamaterial (MM) dual-band THz sensor for various biomedical sensing applications. An MM is a material engineered to have a particular property that is rarely observed in naturally occurring materials with an aperiodic subwavelength arrangement. MM properties across a wide range of frequencies, like high sensitivity and quality factors, remain challenging to obtain. MM-based sensors are useful for the in vitro, non-destructive testing (NDT) of samples. The challenge lies in designing a narrow band resonator such that higher sensitivities can be achieved, which in turn allow for the sensing of ultra-low quantities. We propose a compact structure, consisting of a basic single-square split ring resonator (SRR) with an integrated inverted Z-shaped unit cell. The projected structure provides dual-band frequencies resonating at 0.75 THz and 1.01 THz with unity absorption at resonant peaks. The proposed structure exhibits a narrow bandwidth of 0.022 THz and 0.036 THz at resonances. The resonant frequency exhibits a shift in response to variations in the refractive index of the surrounding medium. This enables the detection of various biomolecules, including cancer cells, glucose, HIV-1, and M13 viruses. The refractive index varies between 1.35 and 1.40. Furthermore, the sensor is characterized by its performance, with an average sensitivity of 2.075 THz and a quality factor of 24.35, making it suitable for various biomedical sensing applications.

## 1. Introduction

The THz portion of the electromagnetic (EM) spectrum behaves like a bridge between microwaves and the infrared range, providing an advantage in biomedicine, material science, sensing, defence, communications, and so on [1]. The THz range typically refers to frequencies ranging from 0.1 to 10 THz. Researchers are interested in absorbers based on metamaterials for the THz spectrum [2,3,4]. Metamaterials (MMs) are composite man-made periodic arrays of unit cells with unique features like a negative refractive index (n), negative permittivity, and so on [5]. These features do not occur in naturally occurring materials. Because of these non-natural properties, MMs have drawn interest for a range of applications, such as in sensors [6,7], ideal absorbers [8], clocking devices [9], stealth applications, biosensors, and antennas [10,11] with improved performance through features such as miniaturization, and bandwidth intensification. Mostly, these materials are characterized by their subwavelength structures. In subwavelength structures, the individual elements are smaller than the radiation wavelength they are designed to interact with. To develop THz sensors, researchers have concentrated on the following objectives: (i) creating ultra-thin absorbers for packaging [1]; (ii) achieving reconfigurability to lower fabrication costs [12,13]; and (iii) implementing narrow/ultra-narrow band absorbers for electromagnetic shielding, packaging [14,15,16,17] and high sensitivity [18]. Reconfigurability is the feature that can reduce the cost of re-fabricating devices the most. For further developments, researchers strive to minimize the utilization of costly and simple designs for re-configurable properties [12]. It was suggested that a sensor with a high quality (Q) factor (296.3), was created by using more than one split ring resonator [19]. In this case, the refractive indices ranged from 1.00 to 1.10. Although these designs have not been demonstrated for cancer detection, there are designs in the literature that have a higher figure of merit (FOM) and Q-factor than the suggested design. A refractive index sensor was proposed to be developed utilizing four-strips [20]. In this case, the range of the refractive index (n) was 1.0–1.6, though this design has not been demonstrated for cancer detection. MM-based sensors are useful for carrying out non-destructive tests (NDTs) for the in vitro evaluation of samples. The design of the metamaterial response here is critical for achieving higher sensitivity, since a narrow-band absorber with unity absorption is required. This allows for the sensing/detection of ultra-low quantities of samples. In this article, we try to showcase a single THz sensor with a dual-band response for the detection of various malignant cells and viruses like HIV, M13, and glucose. To analyze the suggested design, n was chosen in the range of 1.35 to 1.40 because this range encloses the n of various biomolecule samples, which usually exhibit n within this range. Particularly, the identification and characterization of numerous biological samples, including those with anomalous cellular properties like cancer cells having distinct values of n compared to healthy cells, render this range extremely fascinating [21]. A comparative analysis of various refractive index-based THz metamaterials is presented in the subsequent sections.

## 2. Materials and Methods

### 2.1. Absorber Design

In this study, we propose a compact structure, consisting of a basic single-square split ring resonator (SRR) with an integrated inverted Z-shaped unit cell. The THz dual-band sensor was constructed with a multi-layer, periodically arranged metallic design, a substrate, and a continuous metallic plate in the bottom plane to make the transmission zero. The design consists of a combination of two prototypes. Prototype 1 is a basic single-square SRR that is considered an absorber 1 (Abs-1), which exhibits a resonant frequency at 0.768 THz with 94% absorption. Prototype 2 is an inverted Z-shaped unit cell that is considered an absorber 2 (Abs-2), which provides a resonant frequency at 1.0365 THz with an absorption of 95%. The proposed design offers a dual band of resonant frequencies at 0.75 THz and 1.01 THz with an absorption greater than 98%. The Computer Simulation Technology (CST) Studio suite 2022 was used to simulate the presented dual-band THz sensor. The sensor structure with different views and its dimension details are shown in Figure 1 and Table 1, respectively. Figure 1a offers an anterior view of the suggested absorber’s unit cell configuration. The unit cell is designed with gold material having an electrical conductivity of $4.561\times 10^{7}$ S/m and a thickness of $h_{3}$ = 0.5 μm. To maximize the reflection, there is a metal-like gold coating at the bottom of the design and a substrate of silicon with a permittivity of $\epsilon_{s}$ = 11.9 at the top, with a periodicity of width = length = 50 μm. The periodic arrangement of the suggested design is delineated in Figure 1c. Figure 1b,d demonstrate the proposed absorber’s cross-sectional view and 3D perspective view, respectively.

### 2.2. Evolution of the Absorber

Figure 2 depicts the absorption response during the evolution of the proposed structure. Figure 2a,b illustrate Abs-1 and Abs-2, respectively. The design procedure is the same for absorbers 1 and 2 except for the unit cell shape. Figure 2c is the suggested design (Abs-3) and is a combination of Abs-1 and 2. The expression for calculating the absorption coefficient is A = 1 − R − T, where R stands for reflection, T for transmission, and A for absorption. Resonances are available at 0.768 THz for Abs-1 and 1.0365 THz for Abs-2. The coupling between Abs-1 and Abs-2 perturbs the energy levels of Abs-3, leading to a red-shift of the resonant transitions and the emergence of dual absorption bands centered at 0.75 THz and 1.01 THz. Due to the interaction between Abs-1 and 2, there is perfect absorption at resonant frequencies. The first and second resonant frequencies of the proposed design have narrow bandwidths of 0.022 THz and 0.036 THz, respectively. Evolution performance is shown in Table 2.

## 3. Results

### 3.1. Field Analysis

In this section, we analyze the electric (E) and magnetic (H) field distributions to provide a better understanding of the proposed sensor’s operation. Figure 3 illustrates the distribution of the E-field, H-field, and surface current of the proposed design. Figure 3a,b show the E-field distribution, while Figure 3c,d show the H-field distribution. Here, we observe that there are λ and 3λ/2 resonance modes at 0.75 and 1.01 THz, respectively. Firstly, at the first resonating frequency, the surface current is located at the basic single-square SRR arms. At the second resonating frequency, most of the surface current is concentrated at the center of the inverted “Z”, as evident from Figure 3e,f.

### 3.2. Dependence of Resonance Frequency with Design Parameters

The resonant frequency can be engineered by manipulating the electrical parameters of the structure. A red-shift in the resonance is observed by increasing the basic single-square SRR slit width (g) or the dielectric spacer height ($h_{2}$). Conversely, reducing the distance (i) between the inverted Z arm and the basic single-square SRR induces a blue-shift in the resonant frequency. Figure 4 shows the absorption spectra for various design parameters such as the basic single-square SRR width, slit gap, substrate height, and distance (interaction) between inverted Z arms and a basic single-square SRR.

### 3.3. Refractive Index of Analyte = 1.35–1.40

This section presents the suggested THz sensor sensitivity performance for distinct analytes from n profiles. To evaluate sensor performance, the analyte is loaded as depicted in Figure 1d. For various biomedical samples, n is typically in the range of 1.35 to 1.40. The thickness of the analyte remains constant; however, the analyte’s n is varied between 1.35 and 1.40, since the increase in n exhibits a significant red-shift in the resonant frequencies and a negligible change in the absorption peak at the second resonant peak, as shown in Figure 5a. For comparing the performance of the sensor, FOM is one of the vital metrics and is defined as the ratio of sensitivity (S) of the sensor to the full-width half maxima (FWHM) of the resonant frequency ($f_{r}$); it is represented in the following form of expression:(1)$$\;\mathrm{FOM}=\frac{S}{\mathrm{FWHM}}$$

The proposed structure has an average FOM of 69.16 and 45.61, and an average sensitivity of 2.075 and 2.6 THz/RIU for resonant frequencies 0.75 and 1.01 THz, respectively. Figure 6 shows n versus sensitivity, quality factor, figure of merit, and change in resonant frequency. In Figure 6c, the sensitivity increases consistently from n = 1.35 to 1.40 for both resonances, and at n = 1.39, the second resonance becomes more responsive. Similarly, in Figure 6a,b,d, the quality factor, the merit figure, and the change in the resonant frequency remain the same for both resonances from n = 1.35 to 1.4. As n increases, all parameters, including sensitivity, FOM, and change in resonant frequency, exhibit an upward trend. However, the quality factor remains relatively stable.

## 4. Discussion

### 4.1. Sensing Performance

The most crucial performance metric is sensitivity, defined as the ratio between the change in the refractive index (Δn) of the analyte and the corresponding change in its resonant frequency (Δf); its equation form is *S*=Δf⁄Δn (THz/RIU).

### 4.2. Glucose Detection

Water and water containing 25% glucose have a refractive index of $n_{w}$ ≈ 1.3198 and $n_{g}$ ≈ 1.3594, respectively [22]. $n_{w}$ and $n_{g}$ represent the refractive index values of water and glucose, respectively. Table 3 illustrates the sensing capability of the proposed sensor for glucose detection. The table also suggests that the first resonant frequency of 0.75 THz shifted to about 20 GHz, resulting in a sensitivity of 492 GHz/RIU. Similarly, a high sensitivity of 618 GHz/RIU was recorded at the second resonant frequency of 1.01 THz, corresponding to a 24.5 GHz frequency shift. Figure 5b shows the absorption spectrum for the detection of water and glucose. Table 3 not only highlights the sensitivity performance of the proposed dual-band THz sensor but also presents other key metrics, including changes in resonant frequency, Q-factor, and FOM.

### 4.3. HIV-1 and M13 Virus Detection

#### 4.3.1. HIV-1

The Human Immunodeficiency Virus (HIV), related to the retroviridae virus family, has a distinct reverse transcription mechanism. This process enables the virus to change its genetic material from RNA to DNA, allowing it to integrate into the genome of the host cell and spread [23,24,25]. Acquired immunodeficiency syndrome (AIDS) is caused by a viral infection that compromises immunity and impacts multiple systems. Since the virus can only survive inside a human body, it should be transferred through an infected person’s biological fluids such as breast milk, mucus, blood, vaginal secretions, and anal semen. This virus primarily targets CD4 + T cells, which are contaminated and can result in immunosuppression, life-threatening disease, and even death. Immunocompromised status in HIV patients is assessed using CD4 + T cell counts. The n for HIV is 1.5. Figure 7a indicates that the proposed structure’s resonant frequency is red-shifted by 28.5 GHz and 35 GHz for the first and second resonating frequencies with a sensitivity of 282.17 and 345.53 GHz/RIU, respectively.

#### 4.3.2. M13

Bacteriophages, sometimes referred to as bacterial viruses, represent a different mobile genetic entity that is associated with prokaryotic organisms [23,26,27]. They also serve as key agents of gene transfer, facilitating the sharing of genetic material within and between species, and as a regulator of the bacterial population to kill the host bacteria. This distribution plays a crucial role in shaping the microbial landscape and presence in a diverse range of varied habitats such as the freezing temperatures of arctic regions, scorching deserts, hypersaline environments, dirt, and even in the bodies of other organisms [23,26,27]. Its absorption spectrum, shown in Figure 7b, makes clear that there is a shift of 33 GHz and 40 GHz in the first and second resonating frequencies, respectively. Changes in resonant frequency, sensitivity, quality factor, and FOM are shown in Table 4.

### 4.4. Various Cancer Cells’ Detection

#### 4.4.1. Breast Cancer

Breast cancer is among the most prevalent cancers affecting women globally. Every year, the number of women dying from breast cancer is increasing rapidly [28,29]. Research indicates that early detection of breast cancer leads to less complex but costly treatment [30,31]. One of the most used imaging techniques for the diagnosis of breast cancer is X-ray mammography. However, this procedure should not be used frequently as it harms healthy tissues because it involves ionizing X-rays. Alternatively, ultrasound and Magnetic Resonance Imaging (MRI) are additional techniques for detecting breast cancer. This study proposes a method for distinguishing between healthy and cancerous breast cells using a metamaterial-based THz sensor. The analyte is placed near the sensor, and n values are observed. n of healthy cells is 1.449, while that of cancerous cells is 1.5811. Table 5 indicates 24 GHz and 33 GHz shifts in the first and second resonating frequencies, respectively, with a peak sensitivity of 310 GHz/RIU.

#### 4.4.2. Skin Cancer

Skin cancer is among the deadliest diseases that claim lives. Skin cancer may be successfully detected if it is found early on. It could spread to other bodily parts and become much harder to cure if found later. Currently, many scientists are focusing on skin cancer detection techniques to detect this disease early. Various methods, including thermal imaging, Raman microspectroscopy, etc., have been developed for the detection of skin cancer [32,33]. THz radiation has emerged as a highly promising tool in biomedical fields because of its unique properties like non-ionizing with biological systems. The suggested absorber was inspected for skin cancer cells according to the n profile of the surrounding media. The suggested absorber was inspected for skin cancer disease cells; the findings are shown in Table 5. According to Figure 8b, the absorption coefficient of skin cancer cells decreases considerably at the first resonance compared to the absorption coefficient of healthy cells.

#### 4.4.3. MCF-7

It is an expression line for progesterone, estrogen, and glucocorticoid receptors in humans. It was created in 1970 from the pleural effusion of a 69-year-old Caucasian lady with metastatic breast cancer (adenocarcinoma) by Dr. Soule of the Michigan Cancer Foundation in Detroit. Healthy and malignant cells have n of 1.36 and 1.401, respectively [34]. We placed an analyte with n of 1.401 close to the sensor and found that there was a change in the resonant frequency, which red-shifted by 22.5 GHz and 28.5 GHz for two resonances, respectively, with a sensitivity of 549 GHz/RIU and 670 GHz/RIU.

#### 4.4.4. PC12

It is a cell line that originated from an embryonic neural crest pheochromocytoma of the rat adrenal medulla, comprising a blend of neuroblastic and eosinophilic cells. It has an n value of 1.395 for malignant cells and 1.381 for healthy cells [34]. This work initially considered the analyte’s n to be 1.381, resulting in a 22.5 GHz red-shift of the resonance frequency. Upon further analysis, the analyte’s n was determined to be 1.395, leading to a 28.5 GHz red-shift. The observed sensitivities were 1607 GHz/RIU and 1964 GHz/RIU for bands 1 and 2, respectively. The FOM and Q-factor values are tabulated in Table 6. The methodology for calculating the Q-Factor and sensitivity is provided in the Appendix A.

## 5. Proposed Working Setup

For the present work, there is a requirement of a high Q measurement setup at THz frequencies. Photomixer-based THz frequency domain spectroscopy is the most obvious choice. We propose a terahertz (THz) sensor arrangement for sample analysis, as illustrated in Figure 9. A photomixer generates THz radiation, which is further focused onto the sample using a pair of THz lenses that are part of the THz transmitter (THz Tx). At the center of the arrangement, the sample is placed. Because of the interaction between the THz wave and the sample, their properties change depending on the sample features. The modified signal is collected by the THz receiver (THz Rx). The THz Rx consists of photomixers and THz lenses. Photomixers driven by DFB lasers, which are controlled by a precise Peltier-based temperature controller, used to maintain stable operating conditions for the DFB laser and possibly the sample, ensuring consistent results. frequency of THz. This helps us to achieve a Q-factor as high as ≈$10^{5}$ for spectroscopic measurements. A lock-in amplifier is used to capture the detected signal data, which improves the signal-to-noise ratio for more accurate measurement. The Piezo actuator is used to match the source and detector phase to provide coherent measurements. Polarization maintaining (PM) fiber is used to support the polarization of the light as it passes through. Finally, a computer interfaces with the entire setup, gathering data from components of the arrangement. This setup is well suited for studying the interaction of THz signals with the sample, including changes in refractive index or absorption properties, making it ideal for biological sensing applications.

## 6. Conclusions

The potential of metamaterial sensors to function as superior THz sensors has made them a popular subject of study in recent years. The suggested metamaterial sensor is a combination of two-unit cells: one is a basic single-square SRR, and the other is an inverted “Z”. The suggested sensor has an absorption coefficient of 98% and 99% at 0.75 THz and 1.01 THz, respectively, for its resonant frequencies. The refractive index range for which the sensor is computed is 1.35 to 1.4, which is typical for several biomedical samples. The sensor’s highest sensitivity is 2.75 THz/RIU at n = 1.39. Furthermore, the proposed sensor can be utilized in biomedical sensing applications to detect biomedical cells. The suggested approach has applications in breast, skin, MCF-7, and PC12 cancer detection, as well as in detecting the HIV and M13 viruses and glucose. In our future work, we will investigate the possibility of incorporating machine learning techniques to precisely identify the sample characteristics based on the changes in the surrounding media.

## Figures and Tables

**Figure 1 sensors-25-00232-f001:**
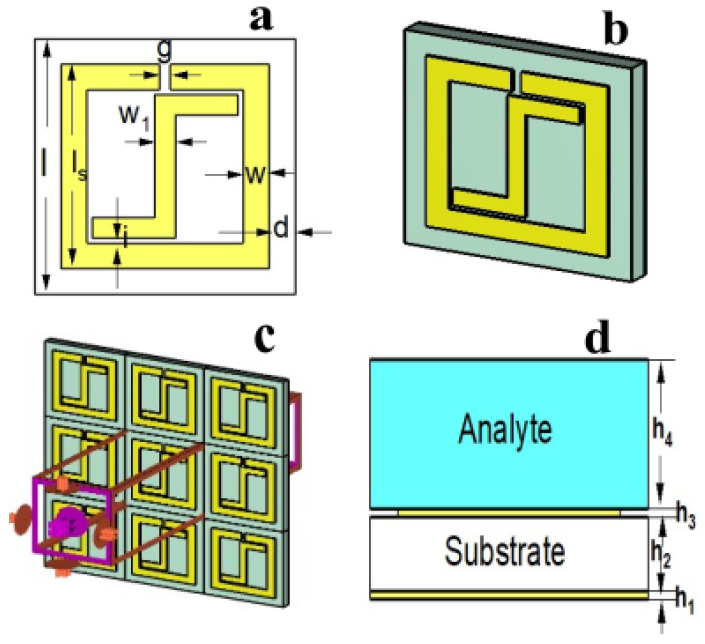
Different outlooks of the sensor structure: (**a**) front view, (**b**) 3D perspective view, (**c**) periodic arrangement, and (**d**) cross-sectional view.

**Figure 2 sensors-25-00232-f002:**
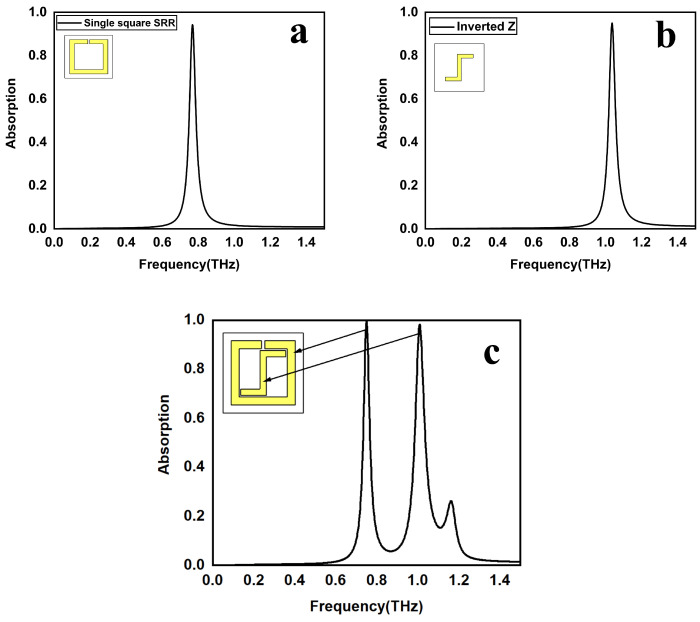
The suggested absorber’s evolution: (**a**) a basic single-square SRR, (**b**) an inverted “Z”-shape, and (**c**) a combination of a basic single-square SRR and inverted Z-shape.

**Figure 3 sensors-25-00232-f003:**
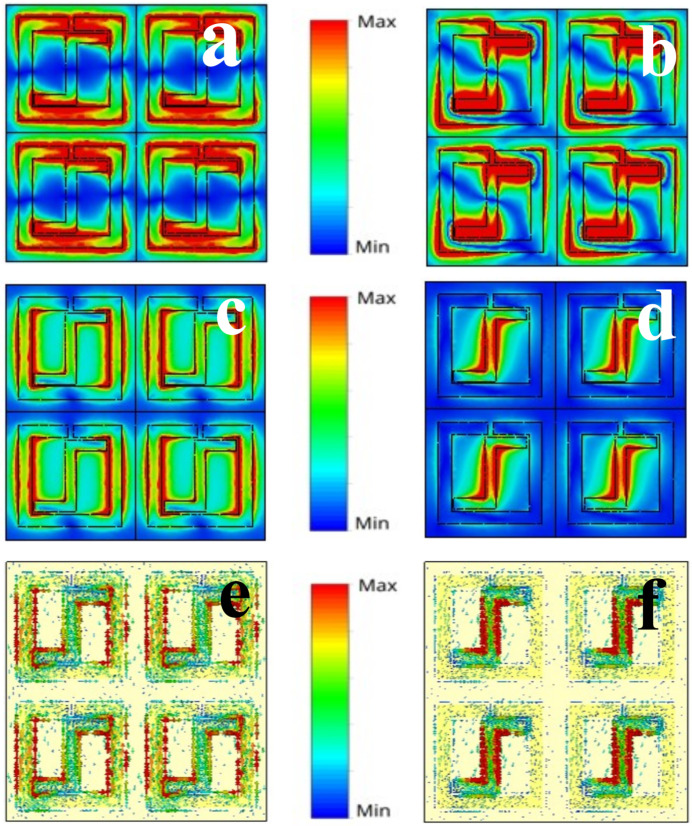
Distribution of electric and magnetic fields, as well as surface currents, at the resonant frequencies of 0.75 THz and 1.01 THz: (**a**,**b**) E-fields, (**c**,**d**) H-fields, and (**e**,**f**) surface current.

**Figure 4 sensors-25-00232-f004:**
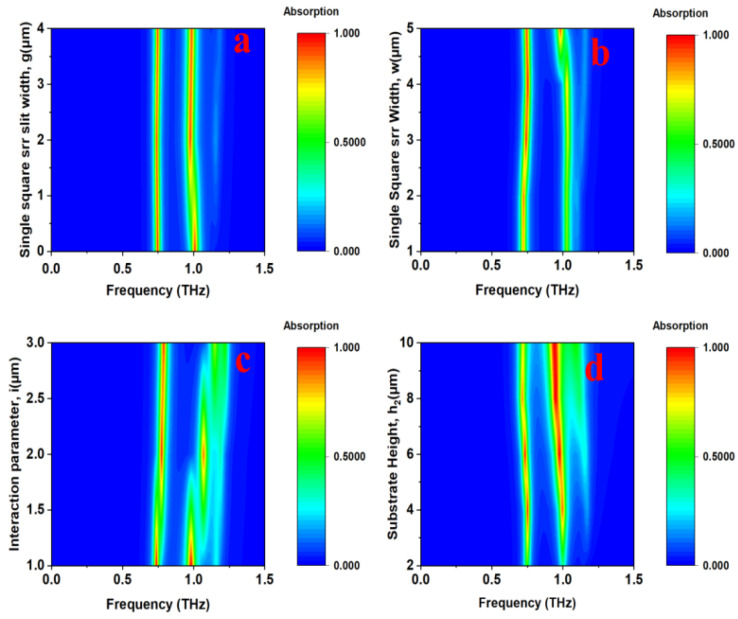
Absorption spectrum analysis by adjusting various design structure parameters: (**a**) slit width (g) of the basic single-square SRR, (**b**) width (w) of the basic single-square SRR, (**c**) interaction (i) between the inverted Z arms and the basic single-square SRR, and (**d**) substrate height $\left(h_{2}\right)$.

**Figure 5 sensors-25-00232-f005:**
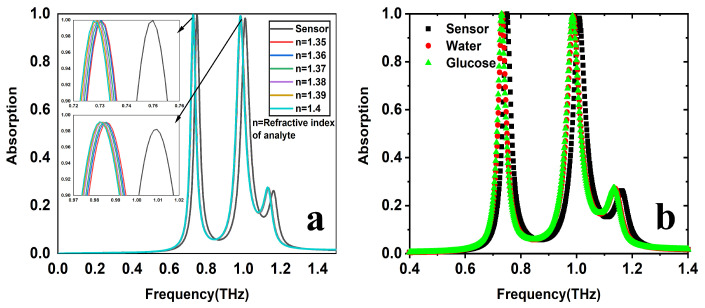
The sensing performance of the sensor for (**a**) analyte n = 1.35 to 1.40 and (**b**) water and glucose detection.

**Figure 6 sensors-25-00232-f006:**
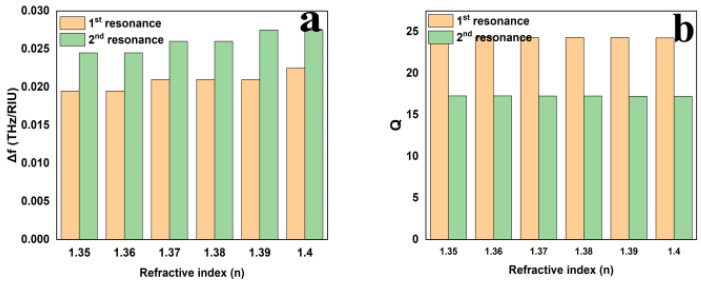
Relationship between the refractive index and (**a**) change in resonant frequency, (**b**) quality factor, (**c**) sensitivity, and (**d**) figure of merit.

**Figure 7 sensors-25-00232-f007:**
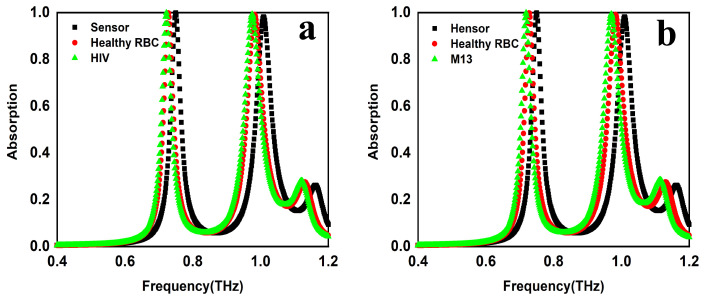
The sensing performance of the sensor for (**a**) HIV virus detection and (**b**) M13 virus detection.

**Figure 8 sensors-25-00232-f008:**
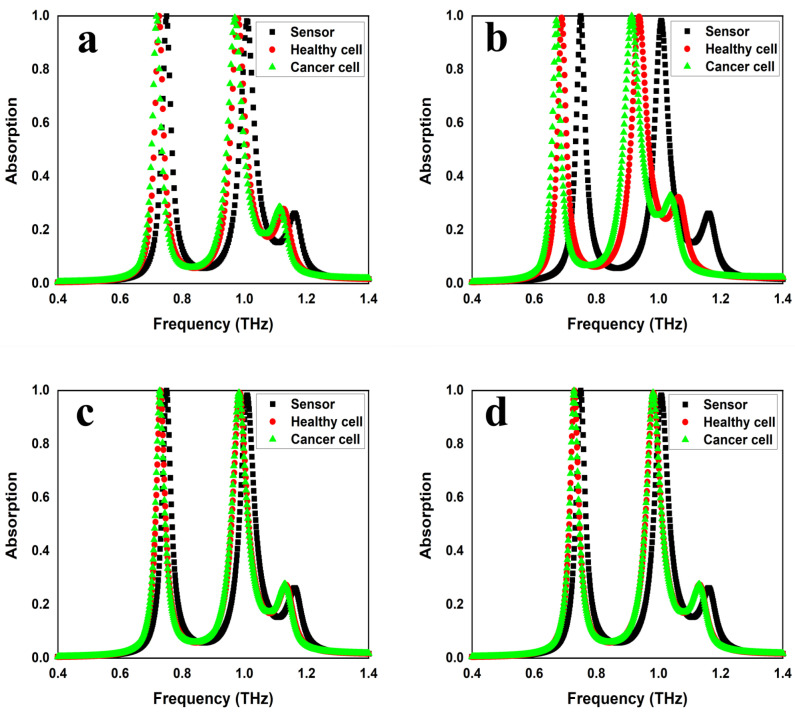
The sensing performance of the design: (**a**) breast cancer detection, (**b**) skin cancer detection, (**c**) MCF-7 detection, and (**d**) PC12 detection.

**Figure 9 sensors-25-00232-f009:**
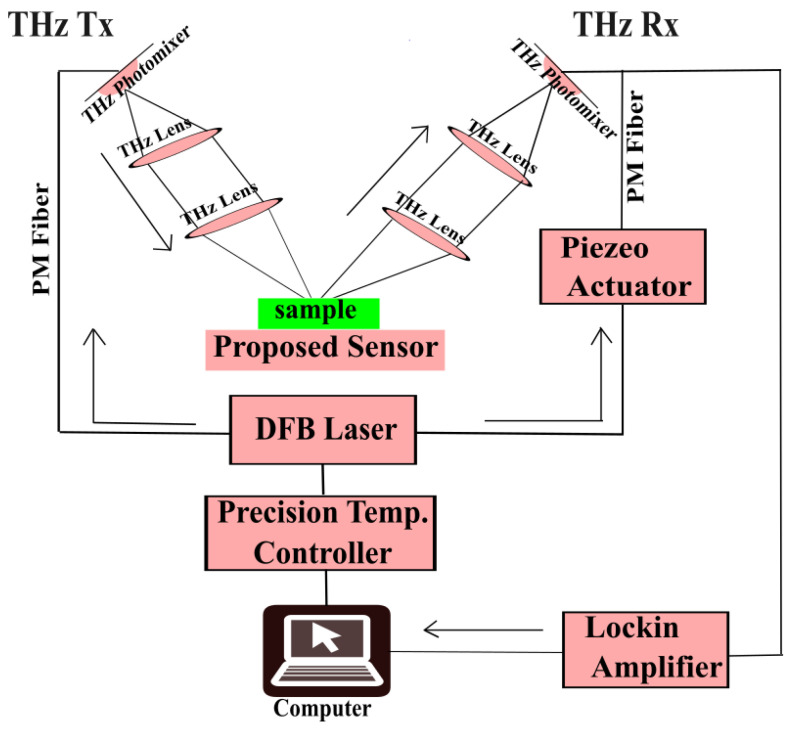
Proposed working setup for the recommended sensor.

**Table 1 sensors-25-00232-t001:** Dimension details of the design.

Parameter	*l*	$l_{s}$	*w*	$w_{1}$	*g*	*i*	*d*	$h_{1}$	$h_{2}$	$h_{3}$	$h_{4}$
value (μm)	50	40	5	4	2	1	5	0.5	5	0.5	10

**Table 2 sensors-25-00232-t002:** Evolution performance.

Absorber	Resonant Frequency (THz)	Absorption (%)	Bandwidth (THz)
Abs-1	$f_{1}$ = 0.768	94	0.025
Abs-2	$f_{1}$ = 1.0365	95	0.033
Abs-3	$f_{1}$ = 0.75	99	0.022
	$f_{2}$ = 1.01	98	0.036

**Table 3 sensors-25-00232-t003:** Water and glucose detection.

Target Substance	n	Band	$f_{r}$ (THz)	Δf (THz)	Q	S (GHz/RIU)	FOM (RIU^−1^)
Water	1.3198	1	0.732	0.018	21.53	56.28	1.6
		2	0.987	0.023	17.94	72	1.3
Glucose	1.3594	1	0.7305	0.02	21.48	492	14.48
		2	0.9855	0.0245	17.23	618	10.85

**Table 4 sensors-25-00232-t004:** HIV and Glucose detection.

Virus	n	Band	$f_{r}$ (THz)	Δf (THz)	Q	S (GHz/RIU)	FOM (RIU^−1^)
HIV	1.5	1	0.7215	0.0285	21.22	282.17	8.3
		2	0.975	0.035	17.10	346.53	6.1
M13	1.57	1	0.717	0.033	21.72	193	6.4
		2	0.97	0.04	17.01	233.92	4.1

**Table 5 sensors-25-00232-t005:** The proposed sensor’s performance in sensing different types of cancer cells.

Cell Name	Analyte Type	n	1st Resonance					2nd Resonance				
			$f_{r}$ (THz)	S (GHz/RIU)	Δf (THz)	Q	FOM (RIU^−1^)	$f_{r}$ (THz)	S (GHz/RIU)	Δf (THz)	Q	FOM (RIU^−1^)
Breast	Healthy	1.449	0.726	250	0.024	24.20	8.3	0.981	310	0.029	17.20	5.44
	Cancer	1.5811	0.717		0.033	23.90		0.969		0.041	17.00	
Skin	Healthy	1.8439	0.6885	380	0.0615	22.95	12.6	0.939	470	0.071	16.50	8.25
	Cancer	2.049	0.672		0.078	22.35		0.9135		0.0965	16.02	
MCF-7	Healthy	1.36	0.7305	549	0.0195	24.35	18.29	0.9845	670	0.0255	17.29	11.75
	Cancer	1.401	0.7275		0.0225	24.25		0.9825		0.0275	17.23	
PC12	Healthy	1.381	0.729	1607	0.021	24.30	53.57	0.984	1964	0.026	17.26	34.46
	Cancer	1.395	0.7275		0.0225	24.25		0.9825		0.0275	17.23	

**Table 6 sensors-25-00232-t006:** Summarizes the comparison between the present work and existing studies.

Ref. No	Size	$f_{r}$ (THz)	Range of n	Step Size of n	S (GHz/RIU)	Q	FOM (RIU^−1^)	Cancer Detection
[35]	120 × 120	1.267	1.0–2.0	0.2	-	40	1.5	No
[36]	36 × 36	2.249	1.35–1.39	0.01	319.6	22.1	2.94	No
[37]	65 × 65	3, 6	1.0–1.8	0.1	834	40.1	11.75	No
[38]	84 × 84	0.1, 1.9	1.4–2.6	0.4	285	32	9.02	No
[39]	36 × 36	5.92	1.1–1.4	0.1	1800	49.6	15	No
[40]	160 × 160	1.931	1.31–1.39	0.01	1045	178.7	-	No
[41]	40 × 40	1.09, 2.8, 4.06	1.0–2.45	-	514.28	13.89	2.257	Yes
This work	50 × 50	0.75, 1.01	1.35–1.4	0.01	2750	25	45	Yes

## Data Availability

The data are available on reasonable request from the corresponding author.

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
