# Peer review of "A Refractive Index-Based Dual-Band Metamaterial Sensor Design and Analysis for Biomedical Sensing Applications"

_sensors, 2025, doi:10.3390/s25010232_

Round 1
Reviewer 1 Report
Comments and Suggestions for Authors
Please see attachment.

Author Response
We thank the reviewer for giving useful feedback. Here we attached one file, file contains the answers for the reviewer comments.

Reviewer 2 Report
Comments and Suggestions for Authors
This manuscript is devoted to development of metamaterial dual-band THz sensor for various biomedical sensing applications. The results of simulation of geometry of sensors and its characteristics are presented. The topic of manuscript is important for scientific and technical groups in areas of THz spectroscopy and its biological applications.
There are some points to make the information more clear or to correct some details:
1) Some of abbreviation were entered some times (e.g. split ring resonator (SRR) was entered in 60th and 101st lines besides of abstract; refractive index (RI or n) was entered in 44th, 155th lines besides of abstract). The abbreviations should be used in similar writing in all text (e.g. the abbreviations for absorber 1 and 2 and combination of them were entered as Abs-1 (64th line), Abs-2 (66th line) and Abs-3 (82nd line) respectively, but further they are used as abs-1, abs-2 and abs-3 (with small letter).
2) It is necessary to add the value of h4 in Table 1. There are four parameters h in Figure 1b, but three of them only are presented in Table 1.
3) It is necessary to arrange the reference numbers in the order in which they are mentioned in the text. Currently, they are cited as 1-7, 15, 9-11, 13-21, 42, 37, 22-27, 38-45, 30-36 (this list is concerned to the first mention of reference). Besides that some of references presented in the bibliography are not cited in text (e.g., 8, 12, 28, 29, 46-51).
4) In addition it is necessary to check the citing, because there are some errors there (e.g. ref 42. It is written “Xie, Qin and Dong, Guang-Xi and Wang, Ben-Xin and Huang, Wei-Qing, High-Q Fano resonance in terahertz frequency based on an asymmetric metamaterial resonator, Nanoscale research letters, vol 13, pp 1–7, 20018”, but it s should be written “Xie, Qin and Dong, Guang-Xi and Wang, Ben-Xin and Huang, Wei-Qing, High-Q Fano resonance in terahertz frequency based on an asymmetric metamaterial resonator, Nanoscale research letters, vol 13, pp 294 (1–7), 2018”). Besides that, in the text authors write “Similarly, a refractive index sensor with a sensitivity of 2.6 THz was proposed to be developed utilizing four-strips [42]. In this case, the range of refractive index (RI) is 1.0-1.6 and this design has not been demonstrated for cancer detection”. (42nd -44th lines). That work [42] is devoted the structures based on planar metamaterial, 1) formed by four-strip metallic resonators, and supporting a two sharp Fano resonances at 0.43THz and 0.81 THz and 2) formed by five-strip metallic resonators, and supporting two sharp Fano dips at 0.75 THz and 0.91 THz. There are no any mentions about 2.6 THz. In addition the phrase of “a sensitivity of 2.6 THz” is not correct. The 2.6 THz is value of frequency, but it is not sensitivity. A sensor's sensitivity indicates how much its output changes when the input quantity it measures changes. Therefore sensitivity is determined by minimal value of parameter or its minimal changes which can be measured. In ref [42] two determination of sensitivity are given: 1) “The sensing sensitivity S is equal to Δf/Δn. Here, S of the sensor is calculated to be 0.105 THz/RIU (refractive index unit)” and in this case “FOM = S /linewidth” and 2) “The sensing capability is also usually discussed by FOM* = S/I and S* = ΔI/Δn , which is related to the detected intensity. The calculation result of S* in this structure is 2.6/RIU”. The value of “2.6” refers to intensity changes but not to frequency changes.
5) Authors write about sensitivity at calculation of FOM in 116th line. But the abovementioned definition of sensitivity through the resonant frequency shift at refraction index unit is written by authors in 131st line. In addition authors ignore the refraction index units in sensitivity units in all text of manuscript. In connection with this definition of sensitivity, Figures 6 a and c should be redone, since for the first value of the refractive index range (n=1,35) there is no change in the refractive index and there cannot be a change in frequency. The resonant frequency value is only measured (calculated). And for the next value of the refractive index (n=1.36) there is already a change (Dn=0.01) and there may already be a change in frequency, which is given. In other case it is necessary to give the initial refractive index and the initial frequency, relative to which these changes are given. Judging by the given figures, since the values of the first and second values are the same, they are made relative to the first point, and, therefore, values should not be given for it (for first point). It is necessary to indicate the correct units (THz/RIU) along the ordinate axis in Fig. 6c.
6) In connection with the above-mentioned definition of sensitivity, the values given in Table 6 remain unclear. It is unclear relative to what initial values the change refractive index occurs. So far it looks like when calculating the sensitivity value for water (454GHz/RIU), the change in the refractive index relative to that for glucose (Dn=Df/S=0.018 THz/(0,454THz/RIU)=0.0396 and nwater+Dn=1.3198+0.0396=1.3594= nglucose) was used.
In the opposite direction for glucose, the values are calculated relative to water. The difference is due to the difference in frequency shifts. Based on this, it remains unclear why different frequency shifts were obtained for the two substances relative to each other. In general, the table requires some explanation.
7) Also Table 4 requires some explanation. If a refractive index of 1.5 is selected, then from considerations similar to the previous point, this sensitivity (282 GHz/RIU) corresponds to a change in the refractive index by 0.1. This means that the change in the refractive index is considered relative to the value n = 1.4. But for this refractive index there is a frequency shift relative to the original n = 1.35 (Fig. 4a). Why is the frequency value relative to which the change is considered selected 0.75 THz (0.7215THz+0.0285THz=0.75THz), and not taking this shift into account?
8) Similar explanations (concerning the initial frequency) are required for Table 5.
9) The first table in text is Table 6. It is necessary to either rewrite the text so that this mention about refractive indices will be without a table but with corresponding references, or to create the separate table 1, or something else.
10) It will be better to use the symbol of “*” or “´” as multiplting symbol, but not the italic letter of “x” (e.g. “4.561x107S/m” (72nd line).
11) Authors write “Here, we observed that there are λ and 3λ resonance modes at 0.75 and 1.01THz respectively” (94th- 95th lines) If that means the first and third harmonics in terms of wavelengths , it is not clear why do authors say about the λ and 3λ modes at these frequencies? The wavelengths of these frequencies (0.75THz and 1.01 THz) are λ =c/f-= 3*1010 (cm/c)/(0.75*1012 (c-1)) =0.04 cm=0.4mm and λ =c/f-= 3*1010 (cm/c)/(1.01*1012 (c-1)) =0.029 cm=0.29mm, respectively. These wavelengths do not have a 1 to 3 ratio to each other. It is necessary to clear what the authors mean.
12) Authors write “To assess the sensor’s performance, load the analyte, as depicted in Fig. 1 (c)” (110th -111th lines) but the presence of the analyte is demonstrated in Figure 1b. In addition the parts of Figure 1 should be designated without the abbreviation ”Fig” but only with letters as in other Figures of manuscript.
The article requires revision for a clearer understanding of the results obtained by the authors and, possibly, some clarifications in the results obtained.
The manuscript can be published after major revisions.
Author Response
We thank the reviewer for giving the informative feedback. Answers for comments are attached.

Reviewer 3 Report
Comments and Suggestions for Authors
The submitted work studies theoretically the properties of a new sensor on the basis of a meta-material structure. My comments thereabout follow:
1. The Authors need to provide the full configuration of detection with the proposed sensor. What are the necessary parameters of the radiation source and the detection apparatus? At least some estimates are necessary.
2. The proposed sensor should be compared to other sensors and/or detection methods. What is the advantage? Is it better sensitivity, lower cost, or simpler method, what is it exactly?
3. The work is rather theoretical and predictive. What kind of problems in practical implementation do the Authors foresee? This should be analysed.
If the Authors take into consideration the above-listed comments, their manuscript may be published in Sensors.
Author Response
We, thank the reviewer for useful feedback. Answers for comments are attached.

Round 2
Reviewer 1 Report
Comments and Suggestions for Authors The authors have carefully addressed all my comments and adequatly revised the manuscript.
Reviewer 3 Report
Comments and Suggestions for Authors
In response to my observations, important information was added to the manuscript that made it more interesting and comprehensible. My comments have been fully addressed by the Authors in the revised manuscript, which may be now published.